# Identification of triciribine as a novel myeloid cell differentiation inducer

**Souma Suzuki[1], Susumu Suzuki[1,2], Yuri Sato-Nagaoka[2], Chisaki Ito[2], Shinichiro Takahashi** [ID][1,2]*

1 Faculty of Medicine, Division of Laboratory Medicine, Tohoku Medical and Pharmaceutical University, Sendai, Miyagi, Japan, 2 Department of Clinical Laboratory, Tohoku Medical and Pharmaceutical University Hospital, Sendai, Miyagi, Japan

* shintakahashi@tohoku-mpu.ac.jp

**Data Availability Statement:** All relevant data are within the article and its Supporting Information files. The data for microarray analysis are openly available in the NCBI Gene Expression Omnibus (http://www.ncbi.nlm.nih.gov/geo/) at the GEO

## Abstract

Differentiation therapy using all-trans retinoic acid (ATRA) for acute promyelocytic leukemia (APL) is well established. However, because the narrow application and tolerance development of ATRA need to be improved, we searched for another efficient myeloid differentiation inducer. Kinase activation is involved in leukemia biology and differentiation block. To identify novel myeloid differentiation inducers, we used a Kinase Inhibitor Screening Library. Using a nitroblue tetrazolium dye reduction assay and real-time quantitative PCR using NB4 APL cells, we revealed that, PD169316, SB203580, SB202190 (p38 MAPK inhibitor), and triciribine (TCN) (Akt inhibitor) potently increased the expression of CD11b. We focused on TCN because it was reported to be well tolerated by patients with advanced hematological malignancies. Nuclear/cytoplasmic (N/C) ratio was significantly decreased, and myelomonocytic markers (*CD11b* and *CD11c*) were potently induced by TCN in both NB4 and acute myeloid leukemia (AML) M2 derived HL-60 cells. Western blot analysis using NB4 cells demonstrated that TCN promoted ERK1/2 phosphorylation, whereas p38 MAPK phosphorylation was not affected, suggesting that activation of the ERK pathway is involved in TCN-induced differentiation. We further examined that whether ATRA may affect phosphorylation of ERK and p38, and found that there was no obvious effect, suggesting that ATRA induced differentiation is different from TCN effect. To reveal the molecular mechanisms involved in TCN-induced differentiation, we performed microarray analysis. Pathway analysis using DAVID software indicated that "hematopoietic cell lineage" and "cytokine-cytokine receptor interaction" pathways were enriched with high significance. Real-time PCR analysis demonstrated that components of these pathways including IL1β, CD3D, IL5RA, ITGA6, CD44, ITGA2B, CD37, CD9, CSF2RA, and IL3RA, were upregulated by TCN-induced differentiation. Collectively, we identified TCN as a novel myeloid cell differentiation inducer, and trials of TCN for APL and non-APL leukemia are worthy of exploration in the future.

## Introduction

Acute myeloid leukemia (AML) is a fatal disease characterized by the proliferation of clonal and abnormally differentiated cells of the hematopoietic system, which infiltrate into the bone

series accession number GSE235632, as well as in the supplementary material of this article.

**Funding:** This work was supported in part by a Grant-in-Aid for Scientific Research (21K07346) from the Ministry of Education, Science and Culture, Japan to ST, Kyowa-Kirin Research Support (Kyowa-Kirin Co. Ltd.) to ST, and Daiichi-Sankyo Research Support (Daiichi-Sankyo Inc.) to ST. The funders had no role in study design, data collection and analysis, decision to publish, or preparation of the manuscript.

**Competing interests:** The authors have declared that no competing interests exist.

marrow, peripheral blood, and other tissues [1]. Cytarabine-based standard chemotherapy is widely used to treat all types of AML, although it can cause severe side effects. All-trans retinoic acid (ATRA) is well established as a differentiation therapy because it is very effective and has few side effects for a specific type of AML, acute promyelocytic leukemia (APL). However, the development of tolerance and narrow application of ATRA need to be improved. The activation of kinase signal transduction pathways contribute to the biology of AML through aberrant cell proliferation and differentiation blockade [2]. Furthermore, kinase inhibitors have been proposed as differentiation agents for acute leukemia [3,4]. Therefore, targeting these signaling pathways might be useful for the molecular-targeted treatment of AML.

Many kinase inhibitors have been reported to induce differentiation, including cyclin-dependent kinase inhibitors, glycogen synthesis kinase 3 inhibitors, Akt inhibitors, p38 mitogen-activated protein kinase (MAPK) inhibitors [5]. In the current study, we identified novel myeloid differentiation inducers using a Kinase Inhibitor Screening Library and found that triciribine (TCN) was an effective myeloid differentiation inducer.

TCN is a tricyclic nucleoside, whose synthesis was first described in 1971 by Shram and Townsen [6]. Subsequent studies indicated that TCN inhibits DNA synthesis, although the precise mechanisms involved remain unclear [7]. Wotring *et al.* [8] reported TCN might inhibit Okazaki fragment initiation and/or DNA polymerase delta. Thereafter, it was revealed that TCN had activity against the Akt enzyme, and had higher selectivity for Akt compared with other kinases, including phosphoinositide 3-kinase (PI3K), protein kinase C, protein kinase A, and extracellular signal-regulated kinase (ERK) 1/2 [9,10]. We revealed that the addition of TCN to APL NB4 cells led to the induction of "hematopoietic cell lineage" and "cytokine-cytokine receptor interaction" pathways, which might activate the ERK/MAPK pathway leading to myeloid differentiation.

## Materials and methods

### Cell culture and reagents

NB4 and HL-60 cells were cultured in RPMI medium containing 10% heat-inactivated fetal bovine serum under 5% $CO_2$ at 37˚C in a humidified atmosphere. Cell cultures were performed using an authenticated technique (https://www.sigmaaldrich.com/japan/ordering/technical-service/recipe-cc.html). The following kinase inhibitors were used for flow cytometry (FCM), morphological analysis, and western blotting in this study: Kinase Inhibitor Screening Library (Cayman, MI, USA), TCN (Cayman), and U0126 (Cayman).

### Nitroblue tetrazolium reduction test (NBT)

For the NBT reduction test, $5 \times 10^5$ cells were incubated in 0.5 ml of a freshly prepared solution containing PBS, NBT (Sigma; 1 tablet/10 ml of PBS), and 0.33 μM phorbol myristate acetate for 30 min at 37˚C. After the blind labeling of each sample, at least 200 cells were counted and the percentage of NBT-positive cells was calculated.

### Surface marker expression analysis by FCM

For FCM analysis, approximately $2–3 \times 10^5$ cells were washed with PBS, and 30 μl aliquots of the cell suspensions were protected from light and incubated with 1 μl of a phycoerythrin (PE)-conjugated mouse anti-human CD11b antibody (BioLegend, San Diego, CA) for 30 min at room temperature. An isotype-matched PE-conjugated mouse IgG (BioLegend) antibody was used as a negative control. After incubation, the samples were analyzed on an LSR Fortessa X-20 Flow cytometer (BD Biosciences, San Jose, CA).

## Morphological analysis

In each experiment, NB4 and HL-60 cells in the logarithmic growth phase were seeded at $2\times10^5$ cells/ml, and induced to differentiate with or without ATRA or TCN. The differentiated cells were collected for analysis at specified times. Differentiation was evaluated by morphology after Wright–Giemsa staining. After the blind labeling of each sample, the nucleus/cytoplasm ratio of at least 200 cells was objectively evaluated, and the data were analyzed statistically and presented as a violin plot using Prism software (version 9.0; GraphPad Software Inc., La Jolla, CA, USA).

## Western blotting

We used a nuclear extract and cytosol preparation kit (Apro Science, Naruto, Tokushima, Japan) following the manufacturer's protocol. An appropriate amount of 1× phosphatase inhibitor cocktail (Roche, Indianapolis, IN) and 1× protease inhibitor cocktail (Roche)) was added and the protein concentration was measured by a Pierce protein assay kit (Thermo Fisher, Waltham, MA, USA). Aliquots of supernatants containing 20–30 μg of protein were separated and immunoblotted. Image was captured by WSE-6300H-CS LuminoGraph III (ATTO, Tokyo, Japan). Total cellular extracts were prepared and immunoblotted as described. Signaling pathway molecules were examined using anti-phospho-p44/42 MAPK ERK1/2 (Thr202/Tyr204), anti-phospho p38(T180/Y182), and p44/42 MAPK mouse monoclonal antibodies (all from Cell Signaling Technology, Beverly, MA, USA). Anti-p38α/stress-activated protein kinase (SAPK)2α mouse monoclonal antibody was obtained from BD Biosciences. Rabbit monoclonal β–actin (Cell Signaling Technology) was employed to access the equal loading of the protein.

## Microarray and mRNA expression analyses

For RNA preparation for real-time PCR analyses, NB4 cells treated with TCN and their control cells were seeded at a density of 2–3×105 cells/ml. For microarray analyses, total cellular RNA was isolated from controls and cells treated with 10 μM TCN for 72 hours, using an RNA Mini Purification Kit (Qiagen, Miami, FL) according to the manufacturer's protocol. The samples were subjected to microarray analyses using a Filgen Clariom™ D Assay (Filgen, Nagoya, Japan). Data were analyzed by Transcriptome Analysis Console (TAC) Software (Thermo Fisher Scientific). KEGG pathway analysis was performed by the Database for Annotation, Visualization and Integrated Discovery (DAVID) (https://david.ncifcrf.gov). The gene expression datasets were deposited in the NCBI Gene Expression Omnibus (http://www.ncbi.nlm.nih.gov/geo/) and are accessible through the GEO series (accession number GSE235632). For RNA preparation for real-time PCR analyses, cells were seeded at a density of 2–3×10^5 cells/ml and treated with 10 μM or the indicated amount of TCN. The cells were harvested after 72 h, or at the specified times. For mRNA expression analyses, cDNAs were prepared from cells using a ReverTra Ace® qPCR RT Kit (Toyobo, Tokyo, Japan). Quantitative PCR was performed using the Thunderbird SYBR qPCR Mix (Toyobo) according to the manufacturer's protocol, an Opticon Mini Real-time PCR Instrument (Bio-Rad, Hercules, CA), and a CFX Connect Real-Time PCR Detection System (Bio-Rad) as previously described [11]. The sequences of the primers are listed in S1 Table. The thermal cycling conditions for all genes were: 95˚C for 15 sec, 35 cycles of 95˚C for 10 s, 60˚C for 45 s, and 72˚C for 3 min.

## Results

### Identification of TCN as a novel myeloid differentiation inducer

To identify novel myeloid differentiation inducers, we used a Kinase Inhibitor Screening Library. The NBT assay using NB4 cells showed there was an increase in the percentage of NBT-positive cells after incubation with 18 of 152 inhibitors (Fig 1A).

Next, we examined the expressions of *CD11b* gene by real-time quantitative PCR. We found that TCN (Akt inhibitor), SB202190, PD169316, CAY10571, SB203580 (p38 MAPK inhibitor) and several other inhibitors increased the expression of *CD11b* (Fig 1B), and this was confirmed by FCM analysis (Fig 2A). p38 MAPK inhibitors were reported to be involved in myeloid differentiation [5]; therefore, we focused on TCN because it has not been identified as a myeloid differentiation inducer, and is well tolerated in patients with advanced hematological malignancies [12]. We assessed the time-course effect of TCN on NB4 cells. As shown in Fig 2B, maximal CD11b expression was observed 72 hours after incubation with TCN.

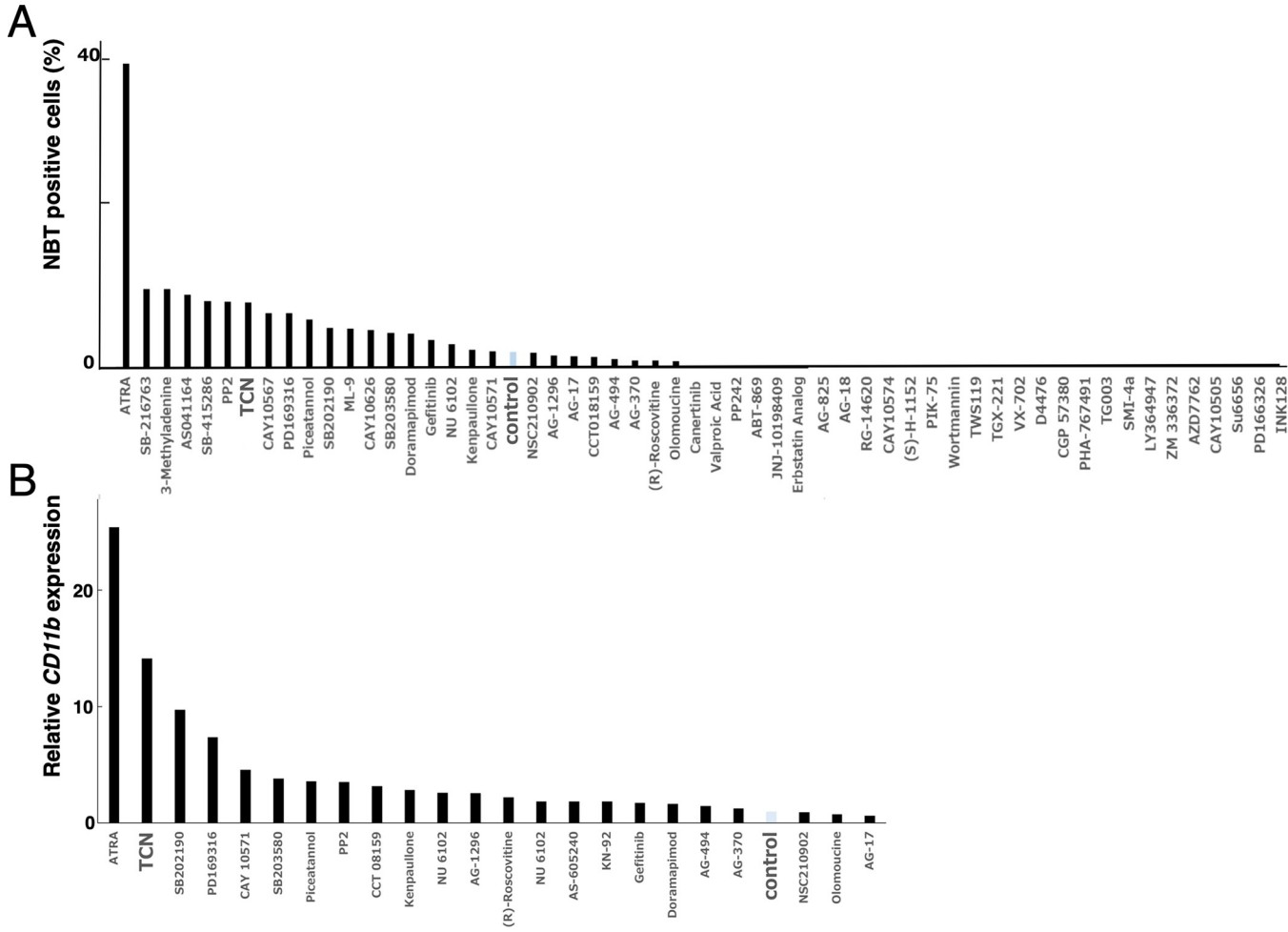

**Fig 1. Identification of TCN as a novel myeloid differentiation inducer.** (A) Effects of ATRA and components of the kinase inhibitor screening library on NBT positive cells. ATRA was used at a final concentration of 100 nM and kinase inhibitors were used at 10 μM, unless noted otherwise. The NBT assay was performed 72 hours after the addition of these agents. (B) Summary of *CD11b* gene expressions by real-time PCR. The relative *CD11b* expression was calculated from the fold expression of *CD11b*, determined by the copy number of *CD11b* divided by the copy number of the internal control, *GAPDH*, compared with control values. The data shown are the mean values of experiments performed in triplicate.

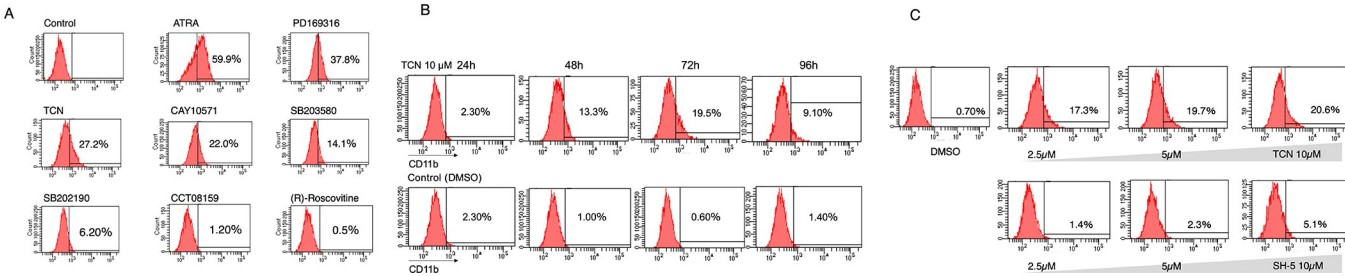

**Fig 2. Effects of ATRA and kinase inhibitors on the expression of CD11b in NB4 cells analyzed by FCM.** (A) The percentages of CD11b positive cells after the addition of the indicated kinase inhibitors are shown as representative histograms of at least three independent experiments. (B) Time course FCM experiment to examine CD11b positive cells after the addition of TCN. Lower panels are controls, to which the solvent dimethylsulfoxide (DMSO) was added. (C) The expression of CD11b in the presence of the indicated doses of TCN and controls (upper panels). The indicated amount of Akt inhibitor SH-5 was added as a comparative experiment (lower panels).

## TCN changes the morphology and potently decreases the nuclear/cytoplasmic ratio of NB4 and HL-60 cells

We next examined the morphology of cells with or without TCN. To prove the efficiency of TCN in inducing cell differentiation in other AML cell types other than NB4, we examined an AML (M2) derived [13] cell line HL-60. As shown in Fig 3A, 72 hours after the addition of TCN, the cytoplasm of NB4 cells was enlarged with slight budding of the plasma membrane region, which was morphologically similar to the effect of 100 nM of ATRA, at the concentration we observed similar nuclear/cytoplasmic (N/C) ratio to 10 μM of TCN (Fig 3B). This indicates that partial myelomonocytic differentiation rather than full myeloid differentiation was achieved [14,15]. Next, we evaluated N/C ratio, which was analyzed statistically and presented as a violin plot. In the absence of TCN, the mean N/C ratio was 70%, which was significantly decreased to 60% in the presence of TCN, which was similar to the effect of the addition of 100 nM ATRA (16) (Fig 3B). In HL-60 cells, the cytoplasm was enlarged like which was seen in NB4 cells (Fig 3C), however, the effect of TCN was even more efficient than 100 nM of ATRA (Fig 3D).

## TCN potently induces myelomonocytic and other differentiation markers in NB4 and HL-60 cells

To characterize the features of TCN-induced NB4 and HL-60 cells, we examined the dose-dependent effect of TCN on several differentiation markers including *CD11b*, *CD11c*, *CD14*, *and myeloperoxidase (MPO)* genes. As shown in Fig 4A–4D, in NB4 cells, real-time PCR analyses demonstrated that TCN potently increased the expressions of myelomonocytic markers *CD11b* and *CD11c* in a dose-dependent manner. In contrast, monocytic *CD14* expression was highest at 2.5 μM TCN and was decreased at a higher concentration (10 μM), whereas *MPO* expression was decreased regardless of the TCN concentration. Similar effect was also observed in HL-60 cells. We next examined the effect of ATRA on *CD11b*, *CD11c*, *CD14*, *and myeloperoxidase (MPO)* genes. Real-time PCR analyses demonstrated that ATRA potently increased the expressions of myelomonocytic markers *CD11b*, *CD11c*, whereas *MPO* expression was decreased, in a dose-dependent manner in both NB4 and HL-60 cells (Fig 4E–4H), which was similar to the TCN effect we observed (Fig 4A–4D). In addition, CD14 expression is induced by ATRA in both lineages.

These results suggest that a high concentration of TCN efficiently induces myelomonocytic markers (*CD11b* and *CD11c*) in NB4 and HL-60 cells.

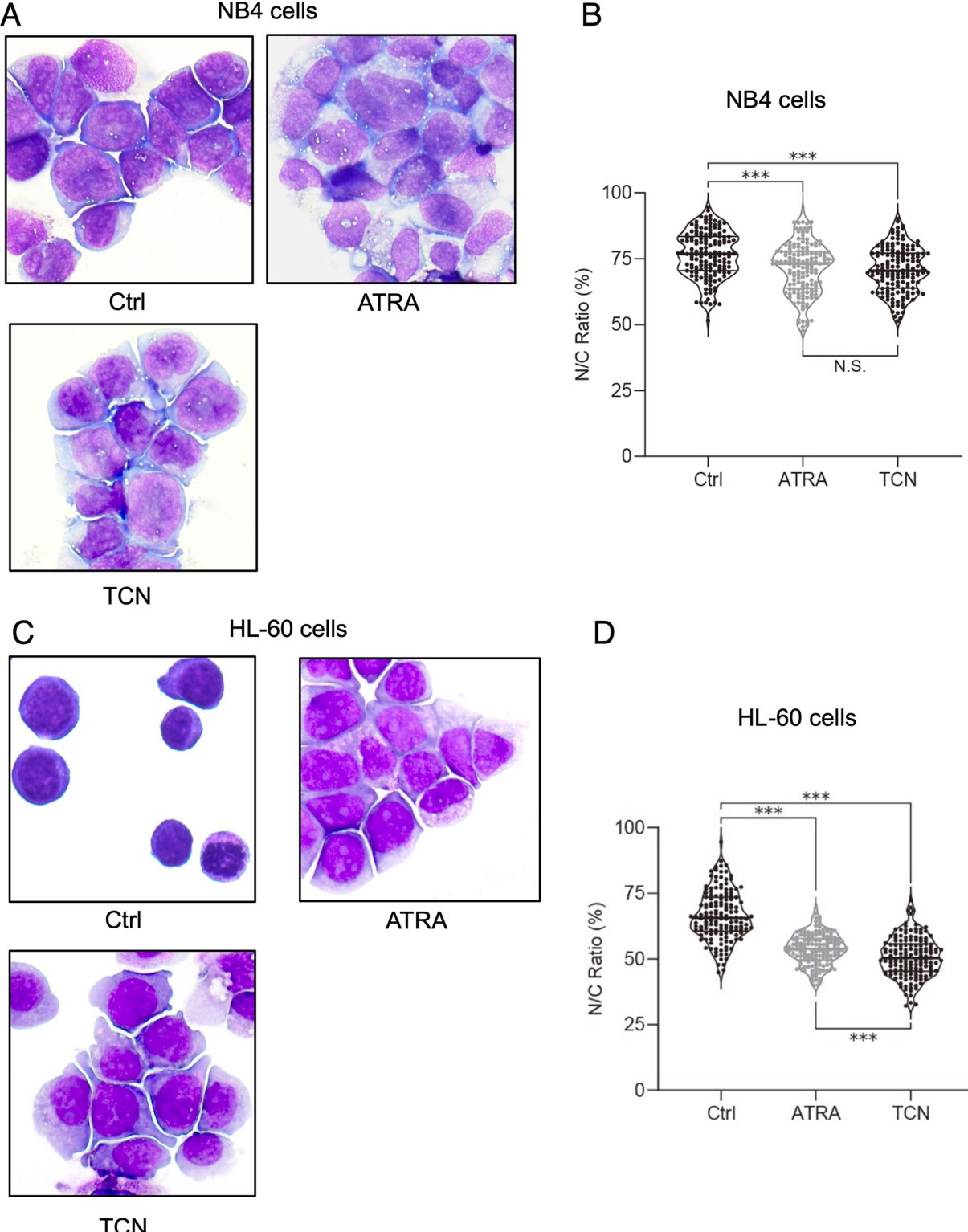

**Fig 3. Morphologic analyses of NB4 and HL-60 cells in the presence or absence of TCN or ATRA.** (A, C) NB4 cells (A) or HL-60 cells (C) were collected and subjected to Wright-Giemsa staining, with and without ATRA (100 nM) or TCN (10 μM) treatment. Magnification = ×400. (B, D) A violin plot of the percentages of the N/C ratio of each cell type, using NB4 cells (B) or HL-60 cells (D). Result of ATRA [16], shown in gray, is presented for comparison (***p<0.001; N.S. not significant).

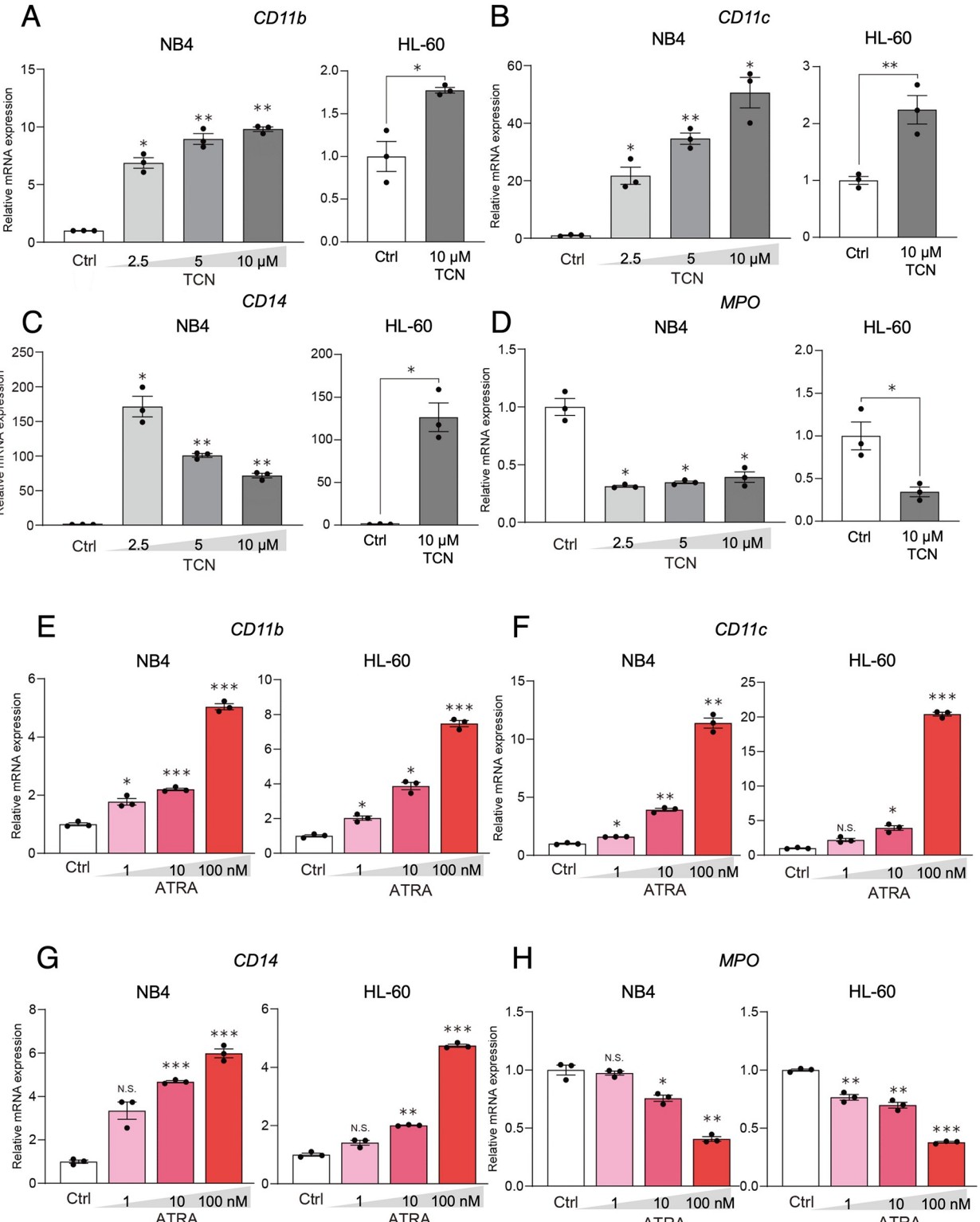

**Fig 4. Analyses of TCN-induced differentiation by differentiation markers.** The TCN-induced differentiation of NB4 and HL-60 cells and control cells was examined by the real-time PCR analysis of several markers. The expressions of (A, E) *CD11b*, (B, F) *CD11c*, (C, G) *CD14*, and (D, H) *MPO*, were examined in the presence of the indicated doses of TCN (A-D) or ATRA (E-H). Relative mRNA expression was calculated from the copy number of the target gene adjusted by *GAPDH*, in each TCN-treated sample (2.5, 5, 10 μM), or ATRA-treated sample (1, 10, 100 nM), which was then divided by the control sample values. The data shown were obtained from three independent PCR amplifications (average ± SD; *p<0.05; **p<0.01; ***p<0.001 vs. Ctrl).

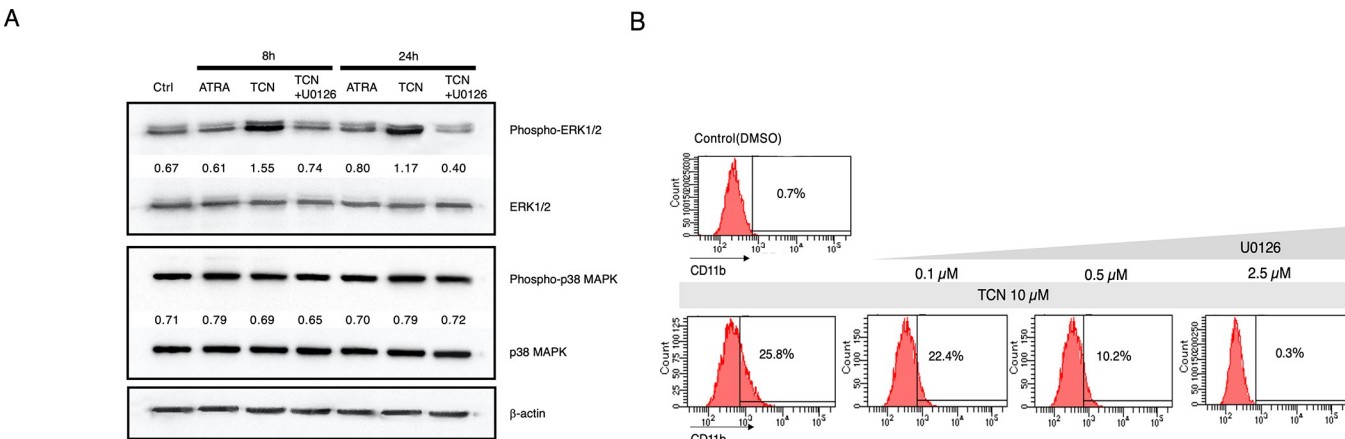

**Fig 5. TCN activates ERK, and the MEK inhibitor U0126 diminishes the induction of CD11b expression by TCN.** (A) Cells were cultured in the presence or absence of TCN or ATRA, for indicated periods, and analyzed for ERK and p38 MAPK activity by western blotting. MEK inhibitor U0126 is also added in the indicated sample. The membranes were probed with an anti-phospho-ERK rabbit polyclonal antibody or anti-phospho-p38 mouse monoclonal antibody. Next, the membranes were stripped and re-probed with an anti-total-ERK mouse monoclonal antibody or anti-total-p38 mouse monoclonal antibody to verify equal protein loading. Indicated numbers are relative density, calculated by Image J 1.54 software, obtained from the amount of density from the band of phospho-ERK, or phosphor-p38, divided by total-ERK, or total-p38. (B) FCM experiment of the effect of TCN on CD11b expression in the presence (0.1, 0.5, 2.5 μM) or absence of the U0126.

## Phosphorylation of ERK is induced by TCN, and the MEK inhibitor U0126 diminishes the induction of CD11b expression on NB4 cells by TCN

We next examined the signaling pathways in NB4 cells affected by TCN. As shown in Fig 5A, Western blot analysis using NB4 cells demonstrated that TCN promoted ERK1/2 phosphorylation, whereas p38 MAPK phosphorylation was not affected, suggesting that activation of the ERK pathway is involved in TCN-induced differentiation. We also examined that whether ATRA may affect phosphorylation of ERK and p38, and found that there was no obvious effect, suggesting that mechanisms of ATRA induced differentiation is different from TCN effect. In addition, TCN induced phosphorylation of ERK was completely abrogated by the addition of MEK inhibitor U0126, suggesting that the involvement of MEK/ERK pathway. We further examined the combination effect of TCN and ATRA for the induction of CD11b expression, and found that there was no enhancement effect (S1 Fig), may also suggest the difference of the mechanisms of these agents. To verify whether the ERK pathway has a role in TCN induced differentiation, we confirmed that the induction of CD11b expression was efficiently reduced by the addition of U0126 (Fig 5B). Collectively, activation of the MEK/ERK pathway is involved in TCN-induced differentiation and this is distinctly different from ATRA induced differentiation.

## Gene expression profiling reveals the effect of TCN on NB4 cell differentiation

To clarify the molecular mechanisms involved in TCN-induced NB4 cell differentiation, we performed microarray analysis. Control NB4 cells and cells incubated with 10 μM TCN were harvested after 72 hours because the maximal effect of TCN was observed at this time point (Fig 2B). When defined to greater than 3-fold changes, TCN upregulated 348 genes and downregulated 83 genes (Fig 6A). We subjected these 348 genes to pathway analysis using DAVID software, and revealed that "hematopoietic cell lineage" and "cytokine-cytokine receptor interaction" were enriched with high significance (Fig 6B). Therefore, we focused on these

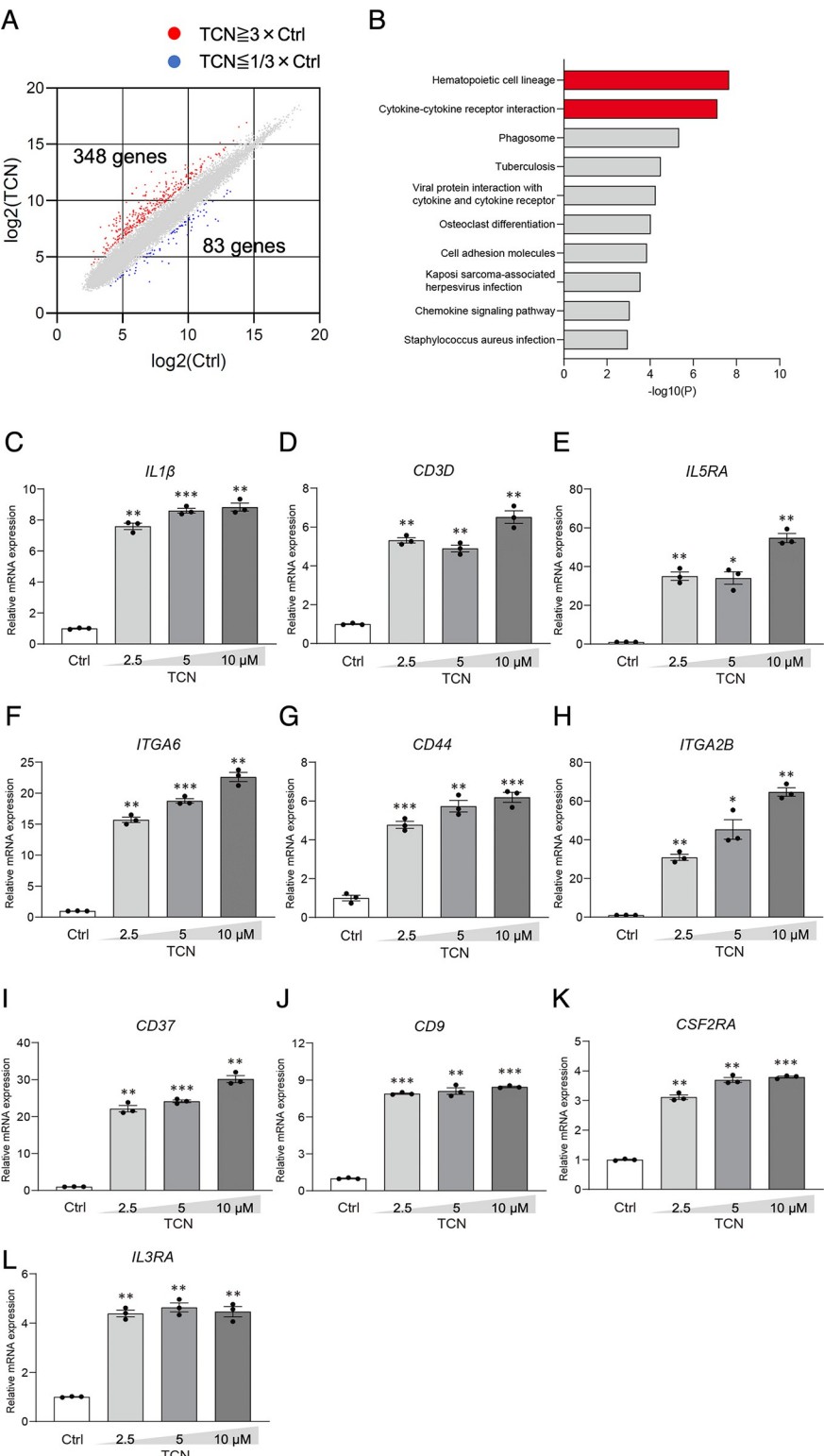

**Fig 6. Results of the microarray analysis.** (A) Gene expression profile of NB4 cells comparing TCN treated vs control cells. (B) Result of pathway analysis using DAVID (https://david.ncifcrf.gov/home.jsp) software from 348 genes. The x-axis shows the negative log base 10 p-value. The higher the number, the higher the significance. (C-L) Total RNAs from NB4 cells cultured in the presence or absence of ATRA and/or TCN for 72 h were analyzed by real-time PCR with specific primers for (C) *IL-1β*, (D) *CD3D*, (E) *IL5RA*, (F) *ITGA6*, (G) *CD44*, (H) *ITGA2B*, (I) *CD37*, (J) *CD9*, (K)

*CSF2RA*, and (L) *IL3RA*. Relative mRNA expressions were calculated from the copy number of the target gene adjusted by *GAPDH*, in each TCN-treated sample (2.5, 5, 10 μM), which were divided by the control sample values. The data presented were obtained from three independent PCR amplifications and reproducibility was confirmed using different batches of cDNA. Statistical significance was determined by one-way ANOVA followed by Dunnett's test. *p<0.05, **p<0.01, ***p<0.001 vs control.

pathways. The results of pathway analysis using the KEGG pathway database and DAVID software are shown in S2 and S3 Figs. A list of the genes related to "hematopoietic cell lineage" and "cytokine-cytokine receptor interaction" are shown in S2 and S3 Tables, respectively. From this list, we selected 11 genes including *CD11b* (S2 and S3 Tables, bold), and verified them by real-time PCR. The other genes were *IL1β, CD3 delta subunit of T-cell receptor complex (CD3D), interleukin 5 receptor subunit alpha (IL5RA), integrin subunit alpha 6 (ITGA6), CD44, integrin subunit alpha 2b (ITGA2B), CD37, CD9, colony stimulating factor 2 receptor subunit alfa (CSF2RA) and interleukin 3 receptor subunit alfa (IL3RA)*. These genes are related to hematopoietic cell development, signal transduction, cell adhesion, immunity, and inflammatory responses. All gene expressions were significantly upregulated by TCN (Fig 6C–6L), suggesting that TCN-induced NB4 cell differentiation is dependent, at least in part, on these genes.

## Discussion

The PI3K/Akt-mammalian target of rapamycin (mTOR) pathway is one of the intracellular pathways aberrantly upregulated in cancers including hematopoietic malignancies [17]. Recently, it was reported that targeting the PI3K/Akt/mTOR signaling pathway had some benefit, even in clinical settings [18–21]. The PI3K/Akt axis is aberrantly activated in T-cell acute lymphoblastic leukemia (T-ALL). The therapeutic potential of TCN for T-ALL was reported by Evangelisti *et al* [22], who showed that TCN, a highly specific Akt inhibitor, induced autophagy and synergized with vincristine, a chemotherapeutic drug for T-ALL patients.

The inhibition of Akt was reported to activate the Raf/MEK/ERK pathway [23]. Bertacchini *et al.* [24]. examined 80 samples of primary cells from AML patients and revealed that the inhibition of Akt and mTOR resulted in the paradoxical activation of growth factor receptor tyrosine kinases (RTKs). In accordance with this, the dual inhibition of RTKs and Akt had synergistic cytotoxic effects in a pre-clinical model [24]. This is consistent with our current result of the activation of ERK pathway by TCN. In the current study, we demonstrated that the signaling effect to ERK, are completely different between TCN and ATRA in NB4 ells. This fact may suggest that synergistic interactions cannot be expected for these two agents. In fact, there was no obvious positive combinational effect for CD11b expression in NB4 cells (data not shown).

In addition, it was previously reported that the MEK/ERK pathway was activated in the early stage of the 1,25-dihydroxyvitamin D-induced monocytic differentiation of HL-60 cells [25]. The Raf/MEK/ERK pathway is considered to control the balance between the expansion and differentiation of hematopoietic progenitors [26,27], and have a critical role during the monocytic and granulocytic differentiation of several myeloid cell lines, including HL-60 and U937 [27]. Therefore, the activation of this pathway by an Akt inhibitor may be critical for myeloid differentiation.

Indeed, many potent myeloid differentiation inducers including ursolic acid [28], Jasmonates, potent lipid regulators in plants [29], *Momordica charantia* seeds [30], Vibsanin A, a novel protein kinase C activator [31], diptoindonesin G (Dip G), a natural resveratrol aneuploid [32], human epidermal growth factor receptor 2 (HER2) inhibitor TAK165 [33], and

staurosporine, a protein kinase C pan-inhibitor [34], were reported to activate the Raf/MEK/ERK pathway. These agents mainly activate ERK/MAPK [28–34] and the downstream STAT1 [32,33], either alone or in combination with ATRA [34], to induce the monocytic and/or granulocytic differentiation of myeloid cell lines including HL-60 [28–32], NB4 [33], U937 [34], or primary AML samples [32].

Inflammatory cytokines, such as IL-1β, tumor necrosis factor (TNF)-α, and the adhesion molecules, L-selectin and intercellular adhesion molecule-1 are upregulated during APL cell differentiation [35,36]. In the current microarray study, we found that the expressions of IL-1β and several integrins were increased (Fig 6), suggesting the TCN-induced differentiation effect may partially overlap with ATRA-induced myeloid differentiation.

Clinical trials of TCN-related agents have been reported in various malignancies including solid tumors [37], metastatic breast cancer [38], and squamous carcinoma of the cervix [39]. However, these trials have reported the limited activity of TCN. Sampath *et al.* [12] reported that TCN phosphate monohydrate therapy was well tolerated in patients with advanced hematological malignancies. In their study, 32 of 41 treated patients were considered evaluable for responses. Among these, 3 patients had decreased marrow blast counts, 1 had a marked reduction in leukocytosis and spleen size, and 17 had stable disease with acceptable toxicities. Because TCN had limited toxicity in these malignancies [12,37–39], basic and clinical research, including combination trials of TCN for APL and non-APL leukemia are worthy of exploration. Further clarification of the role of TCN for myeloid differentiation should be elucidated.

## Supporting information

**S1 Fig. Effect of TCN and ATRA for CD11b expression in NB4 cells, examined by the real-time PCR analysis.** Relative mRNA expression was calculated from the copy number of the target gene adjusted by *GAPDH*, in each TCN-treated sample (10 µM), or ATRA-treated sample (1, 10, 100 nM), which was then divided by the control sample values. The data shown were obtained from three independent PCR amplifications Statistical significance was determined by two-way ANOVA followed by Tukey's multiple comparisons test (*** $p < 0.001$ vs. ctrl, δδ $p < 0.01$, δδδ $p < 0.001$ vs. ATRA).
(TIFF)

**S2 Fig. The pathway analysis of 348 genes in the hematopoietic cell lineage pathway (Fig 6A) using the KEGG pathway database and DAVID software.** Red stars are genes with expressions induced more than 3-fold by TCN compared with controls.
(TIFF)

**S3 Fig. The pathway analysis of 348 genes in the cytokine-cytokine receptor interaction pathway (Fig 6A) using the KEGG pathway database and DAVID software.** Red stars are genes with expressions induced more than 3-fold by TCN compared with controls.
(TIFF)

**S1 Raw images. Western blot raw images for Fig 5A.**
(PDF)

**S1 Table. Sequences of primers used for real-time quantitative PCR.**
(XLSX)

**S2 Table. List of genes in the hematopoietic cell lineage pathway from pathway analysis using the KEGG pathway database and DAVID software.** Bold text indicates genes selected and verified by real-time PCR.
(XLSX)

**S3 Table. List of genes in the cytokine-cytokine receptor interaction pathway from pathway analysis using the KEGG pathway database and DAVID software.** Bold text indicates genes selected and verified by real-time PCR.
(XLSX)

## Acknowledgments

We thank J. Ludovic Croxford, PhD, from Edanz (https://jp.edanz.com/ac) for editing a draft of this manuscript.

## Author Contributions

**Conceptualization:** Shinichiro Takahashi.

**Data curation:** Shinichiro Takahashi.

**Formal analysis:** Susumu Suzuki, Shinichiro Takahashi.

**Funding acquisition:** Shinichiro Takahashi.

**Investigation:** Souma Suzuki, Susumu Suzuki, Yuri Sato-Nagaoka, Chisaki Ito, Shinichiro Takahashi.

**Methodology:** Shinichiro Takahashi.

**Project administration:** Shinichiro Takahashi.

**Resources:** Shinichiro Takahashi.

**Supervision:** Shinichiro Takahashi.

**Validation:** Susumu Suzuki.

**Visualization:** Shinichiro Takahashi.

**Writing – original draft:** Shinichiro Takahashi.

**Writing – review & editing:** Shinichiro Takahashi.

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
