## [Decision Letter · Decision Letter 0]

29 Jan 2024

PONE-D-23-22524Identification of triciribine as a novel myeloid cell differentiation inducerPLOS ONE

Dear Dr. Takahashi,

Thank you for submitting your manuscript to PLOS ONE. After careful consideration, we feel that it has merit but does not fully meet PLOS ONE’s publication criteria as it currently stands. Therefore, we invite you to submit a revised version of the manuscript that addresses the points raised during the review process.

We look forward to receiving your revised manuscript.

Kind regards,

Tanay Chaubal

Academic Editor

PLOS ONE

Journal Requirements:

3. Please upload a copy of Figure 7, to which you refer in your text on page 15. If the figure is no longer to be included as part of the submission please remove all reference to it within the text.

Additional Editor Comments:

** I thank you for submitting your valuable research.**

**Kindly respond to the queries raised by the reviewers. My sincere apologies for the delay in getting a peer review of your research.**

Reviewers' comments:

Reviewer's Responses to Questions

**Comments to the Author**

1. Is the manuscript technically sound, and do the data support the conclusions?

Reviewer #1: Partly

Reviewer #2: Yes

2. Has the statistical analysis been performed appropriately and rigorously? 

Reviewer #1: No

Reviewer #2: Yes

3. Have the authors made all data underlying the findings in their manuscript fully available?

Reviewer #1: Yes

Reviewer #2: Yes

4. Is the manuscript presented in an intelligible fashion and written in standard English?

Reviewer #1: Yes

Reviewer #2: Yes

5. Review Comments to the Author

Reviewer #1: The authors have presented an interesting and clinically important study. However, the following observations have been made considering the study aims to prove that TCN can be used as a clinical alternative to ATRA in the treatment of AML:

The study only compares the effects of ATRA and other kinase inhibitors on NBT positive cells and the expression of CD11b gene.

The study should compare the effect of ATRA to TCN on:

1. The morphology of NB4 cells

2. The induction of myelomonocytic and other differentiation markers in markers in NB4 cells, especially CD11b, CD11c, CD14 and MPO, considering TCN only effectively induces CD11b and CD11c at high concentrations

3. The signaling pathways in NB4 cells

These comparisons would make for a more robust study on the role of TCN in myeloid differentiation and as an effective alternative to ATRA.

Reviewer #2: In this manuscript, Suzuki and colleagues propose the use of Triciribine (TCN), an Akt inhibitor, to potentially improve the treatment of AML because it induces cell differentiation otherwise only previously shown to reduce the growth and survival of cancer cells. TCN has already been tested in a phase I clinical trial for the treatment of advanced hematological malignancies and has been shown to be well tolerated by patients. They used the NB4 cell line to test its cell differentiation efficiency and related pathways activated by the drug in in vitro assays. Although the results were innovative and the experiments were well designed, I have few comments regarding the conceptualization:

1) The authors use the following phrase “because the narrow application and tolerance development of ATRA need to be improved, we searched for another efficient myeloid differentiation inducer” as a rationale for improving the treatment of patients with AML/APL. In this sense, in addition to the NB4 cell line (sensitive to ATRA), they should have tested the efficiency of TNC in cell lines resistant to ATRA. Or prove its efficiency in inducing cell differentiation in other AML cell types other than NB4 (an APL cell line).

2) If they only have access to the NB4 cell line. Why not test drug combinations? Is TNC+ATRA superior in inducing cell differentiation or cell death? It would make more sense, since ATRA (low dose) is still superior to TCN in inducing cell differentiation in APL cells.

3) Line 256: “L) total RNAs from NB4 cells cultured in the presence or absence of ATRA and/or 6AF”. Did the authors want to write TCN instead of 6AF?

6. PLOS authors have the option to publish the peer review history of their article (what does this mean?). If published, this will include your full peer review and any attached files.

Reviewer #1: No

Reviewer #2: **Yes: **Luciana Yamamoto de Almeida

---

## [Author Response · Author response to Decision Letter 0]

27 Feb 2024

Reviewer #1: The authors have presented an interesting and clinically important study. However, the following observations have been made considering the study aims to prove that TCN can be used as a clinical alternative to ATRA in the treatment of AML:

The study only compares the effects of ATRA and other kinase inhibitors on NBT positive cells and the expression of CD11b gene.

The study should compare the effect of ATRA to TCN on:

1. The morphology of NB4 cells

2. The induction of myelomonocytic and other differentiation markers in markers in NB4 cells, especially CD11b, CD11c, CD14 and MPO, considering TCN only effectively induces CD11b and CD11c at high concentrations

3. The signaling pathways in NB4 cells

These comparisons would make for a more robust study on the role of TCN in myeloid differentiation and as an effective alternative to ATRA.

 Thank you very much for these constructive comments. To follow this, we compared the results of the effect of ATRA on the 1. morphology, and 2. differentiation markers, not only in NB4 cells, but in this revised version, we employed AML (M2) derived (13) cell line HL-60 cells. These results are incorporated in the revised manuscript as follows,

Nuclear/cytoplasmic (N/C) ratio was significantly decreased, and myelomonocytic markers (CD11b and CD11c) were potently induced by TCN in both NB4 and acute myeloid leukemia (AML) M2 derived HL-60 cells. (line 25)

To prove the efficiency of TCN in inducing cell differentiation in other AML cell types other than NB4, we examined an AML (M2) derived (13) cell line HL-60. As shown in Fig. 3A, 72 hours after the addition of TCN, the cytoplasm of NB4 cells was enlarged with slight budding of the plasma membrane region, which was morphologically similar to the effect of 100 nM of ATRA, at the concentration we observed similar nuclear/cytoplasmic (N/C) ratio to 10 µM of TCN (Fig 3B). (line 180)

Next, we evaluated N/C ratio, which was analyzed statistically and presented as a violin plot. In the absence of TCN, the mean N/C ratio was 70%, which was significantly decreased to 60% in the presence of TCN, which was similar to the effect of the addition of 100 nM ATRA (16) (Fig. 3B). In HL-60 cells, the cytoplasm was enlarged like which was seen in NB4 cells (Fig. 3C), however, the effect of TCN was even more efficient than 100 nM of ATRA (Fig. 3D). (line 188)

Similar effect was also observed in HL-60 cells. We next examined the effect of ATRA on CD11b, CD11c, CD14, and myeloperoxidase (MPO) genes. Real-time PCR analyses demonstrated that ATRA potently increased the expressions of myelomonocytic markers CD11b, CD11c, whereas MPO expression was decreased, in a dose-dependent manner in both NB4 and HL-60 cells (Fig. 4 E-H), which was similar to the effect of TCN effect we observed (Fig. 4 A-D). In addition, CD14 expression is induced by ATRA in NB4 cells but not in HL-60 cells. (line 211)

Moreover, we examined 3. signaling affected by ATRA and TCN in NB4 cells and the results were incorporated as follows,

We further examined that whether ATRA may affect phosphorylation of ERK and p38, and found that there was no obvious effect compared to TCN, suggesting that ATRA induced differentiation is different from TCN effect. (line 30)

We also examined that whether ATRA may affect phosphorylation of ERK and p38, and found that there was no obvious effect compared to TCN, suggesting that mechanisms of ATRA induced differentiation is different from TCN effect. In addition, TCN induced phosphorylation of ERK was completely abrogated by the addition of MEK inhibitor U0126, suggesting that the involvement of MEK/ERK pathway. We further examined the combination effect of TCN and ATRA for the induction of CD11b expression, and found that there was no enhancement effect (data not shown), may also suggest the difference of the mechanisms of these agents. To verify whether the ERK pathway has a role in TCN induced differentiation, we confirmed that the induction of CD11b expression was efficiently reduced by the addition of U0126 (Fig. 5B). Collectively, activation of the MEK/ERK pathway is involved in TCN-induced differentiation and this is distinctly different from ATRA induced differentiation.

 (line 241)

Additionally, we examined the combination effect of TCN with ATRA in NB4 cells, because other reviewer requested, and found that there was no effect for CD11b induction (as below). This may also suggest that the mechanisms of the effect of TCN is different from ATRA.

I appreciate very much for these constructive comments. Following these comments, I feel that quality of our paper improved greatly. Thank you so much.

Reviewer #2: In this manuscript, Suzuki and colleagues propose the use of Triciribine (TCN), an Akt inhibitor, to potentially improve the treatment of AML because it induces cell differentiation otherwise only previously shown to reduce the growth and survival of cancer cells. TCN has already been tested in a phase I clinical trial for the treatment of advanced hematological malignancies and has been shown to be well tolerated by patients. They used the NB4 cell line to test its cell differentiation efficiency and related pathways activated by the drug in in vitro assays. Although the results were innovative and the experiments were well designed, I have few comments regarding the conceptualization:

1) The authors use the following phrase “because the narrow application and tolerance development of ATRA need to be improved, we searched for another efficient myeloid differentiation inducer” as a rationale for improving the treatment of patients with AML/APL. In this sense, in addition to the NB4 cell line (sensitive to ATRA), they should have tested the efficiency of TNC in cell lines resistant to ATRA. Or prove its efficiency in inducing cell differentiation in other AML cell types other than NB4 (an APL cell line).

Thank you very much for pointing this out. To follow this, in this revised version, we employed AML (M2) derived (13) cell line HL-60 cells, which is other AML cell types than NB4. These results are incorporated in the revised manuscript as follows,

To prove the efficiency of TCN in inducing cell differentiation in other AML cell types other than NB4, we examined an AML (M2) derived (13) cell line HL-60. (line 180)

In HL-60 cells, the cytoplasm was enlarged like which was seen in NB4 cells (Fig. 3C), however, the effect of TCN was even more efficient than 100 nM of ATRA (Fig. 3D). (line 191)

Similar effect was also observed in HL-60 cells. (line 212)

These results suggest that a high concentration of TCN efficiently induces myelomonocytic markers (CD11b and CD11c) in NB4 and HL-60 cells. (line 219)

2) If they only have access to the NB4 cell line. Why not test drug combinations? Is TNC+ATRA superior in inducing cell differentiation or cell death? It would make more sense, since ATRA (low dose) is still superior to TCN in inducing cell differentiation in APL cells.

In this revised version, we examined the effect of TCN not only to NB4 cells but AML M2 derived HL-60 cells, which is other AML cell types than NB4. As a result, we found that TCN is also effective in HL-60 cells as described above.

Moreover, to follow this comment, we examined the combination effect of TCN with ATRA in NB4 cells, and found that there was no effect for CD11b induction (as below). This may probably from the mechanisms for the induction of differentiation by TCN is different from ATRA, in terms of ERK phosphorylation.

These are incorporated in Results, and Discussion, as follows:

We further examined the combination effect of TCN and ATRA for the induction of CD11b expression, and found that there was no enhancement effect (data not shown), may also suggest the difference of the mechanisms of these agents. (line 246)

In the current study, we demonstrated that the signaling effect to ERK, are completely different between TCN and ATRA in NB4 cells. This fact may suggest that synergistic interactions cannot be expected for these two agents. In fact, there was no obvious positive combinational effect for CD11b expression in NB4 cells (data not shown). (line 320)

However, as pointed out, it is very interesting to see the combination effect of ATRA and TCN to other cell lines than APL. It would be grateful if you allow us to move forward this time and would like to examine this as a future project. 

3) Line 256: “L) total RNAs from NB4 cells cultured in the presence or absence of ATRA and/or 6AF”. Did the authors want to write TCN instead of 6AF?

Absolutely, and we sincerely apologize for this mistake. We intended to write TCN, instead of 6AF. Thank you very much for pointing this out.

By following these comments, we feel our paper has improved so much. We really appreciate this.

---

## [Decision Letter · Decision Letter 1]

12 Mar 2024

PONE-D-23-22524R1Identification of triciribine as a novel myeloid cell differentiation inducerPLOS ONE

Dear Dr. Takahashi,

Thank you for submitting your manuscript to PLOS ONE. After careful consideration, we feel that it has merit but does not fully meet PLOS ONE’s publication criteria as it currently stands. Therefore, we invite you to submit a revised version of the manuscript that addresses the points raised during the review process.

**ACADEMIC EDITOR: I thank you for the perseverance and responding to the reviewers' comments. However, there are still some queries which need to be addressed.**

We look forward to receiving your revised manuscript.

Kind regards,

Tanay Chaubal

Academic Editor

PLOS ONE

Journal Requirements:

Reviewers' comments:

Reviewer's Responses to Questions

**Comments to the Author**

1. If the authors have adequately addressed your comments raised in a previous round of review and you feel that this manuscript is now acceptable for publication, you may indicate that here to bypass the “Comments to the Author” section, enter your conflict of interest statement in the “Confidential to Editor” section, and submit your "Accept" recommendation.

Reviewer #1: All comments have been addressed

Reviewer #2: (No Response)

2. Is the manuscript technically sound, and do the data support the conclusions?

Reviewer #1: Yes

Reviewer #2: Partly

3. Has the statistical analysis been performed appropriately and rigorously? 

Reviewer #1: Yes

Reviewer #2: Yes

4. Have the authors made all data underlying the findings in their manuscript fully available?

Reviewer #1: Yes

Reviewer #2: No

5. Is the manuscript presented in an intelligible fashion and written in standard English?

Reviewer #1: Yes

Reviewer #2: Yes

6. Review Comments to the Author

Reviewer #1: (No Response)

Reviewer #2: The authors provided significant improvements to the manuscript. However, I recommend some edits and another experiment to support the data that has been added to this manuscript and increase the relevance of this study.

Abstract: It is missing a conclusion showing the relevance of this study to support a future clinical perspective.

Line 217: Please revise the sentence “which was similar to the effect of TCN effect we observed” and replace it by “which was similar to the TCN effect we observed.”

Lines 217-218: Please correct the sentence “In addition, CD14 expression is induced by ATRA in NB4 cells but not in HL-60 cells.” Figure 4G shows that ATRA induces increased CD14 expression in both cell lineages (NB4 and HL-60).

Do you consider the involvement of the MEK/ERK signaling pathway mediating the cell differentiation process of AML cell lines other than NB4 cells (in this case HL-60)? ATRA is the best inducer of cell differentiation for NB4 cells, but TCN significantly reduced the N/C ratio in HL-60 cells when compared to ATRA (Fig. 3D). Although in the discussion section the authors commented about the relevance of other agents in inducing cell differentiation through the activation of the MEK/ERK pathway in HL-60 cells, I would recommend the authors to perform the same western blot assay (Fig. 5) for HL-60 cells while trying to highlight the novelty of using TCN as a relevant inducer of cell differentiation for other AML types. It would increase the relevance for future research and a potential clinical use of TCN for AML not APL.

Lines 246-248: “We further examined the combination effect of TCN and ATRA for the induction of CD11b expression and found that there was no enhancement effect (data not shown), may also suggest the difference of the mechanisms of these agents.” Why not show the data as supporting information? It is important to show that these drugs with pro-differentiation effects together do not increase cell differentiation in the cell lines tested as an available data. After adding the data, please remove “data not shown” from the discussion section (line 324).

7. PLOS authors have the option to publish the peer review history of their article (what does this mean?). If published, this will include your full peer review and any attached files.

Reviewer #1: No

Reviewer #2: No

---

## [Author Response · Author response to Decision Letter 1]

12 Apr 2024

Reviewer #2: The authors provided significant improvements to the manuscript. However, I recommend some edits and another experiment to support the data that has been added to this manuscript and increase the relevance of this study.

Abstract: It is missing a conclusion showing the relevance of this study to support a future clinical perspective.

 Thank you very much for this constructive comment. To follow this, we added following sentence in abstract (line 38),

Collectively, we identified TCN as a novel myeloid cell differentiation inducer, and trials of TCN for APL and non-APL leukemia are worthy of exploration in the future.

Line 217: Please revise the sentence “which was similar to the effect of TCN effect we observed” and replace it by “which was similar to the TCN effect we observed.”

Thank you for pointing this out. We corrected as follows (line 218),

 which was similar to the TCN effect we observed (Fig. 4A-D).

Lines 217-218: Please correct the sentence “In addition, CD14 expression is induced by ATRA in NB4 cells but not in HL-60 cells.” Figure 4G shows that ATRA induces increased CD14 expression in both cell lineages (NB4 and HL-60).

We apologize for this mistake. We corrected as follows (line 218),

In addition, CD14 expression is induced by ATRA in both lineages.

Do you consider the involvement of the MEK/ERK signaling pathway mediating the cell differentiation process of AML cell lines other than NB4 cells (in this case HL-60)? ATRA is the best inducer of cell differentiation for NB4 cells, but TCN significantly reduced the N/C ratio in HL-60 cells when compared to ATRA (Fig. 3D). Although in the discussion section the authors commented about the relevance of other agents in inducing cell differentiation through the activation of the MEK/ERK pathway in HL-60 cells, I would recommend the authors to perform the same western blot assay (Fig. 5) for HL-60 cells while trying to highlight the novelty of using TCN as a relevant inducer of cell differentiation for other AML types. It would increase the relevance for future research and a potential clinical use of TCN for AML not APL.

Yes, we think MEK/ERK signaling pathway mediates the cell differentiation process by TCN in HL-60 cells as well. However, we don’t think it is good to present HL-60 Western blot in parallel to NB4 (Fig. 5A). We currently found that when combined with ATRA, TCN effect for gene expression is different from NB4 (Fig. S1) to HL-60 cells. If we present the Western blot for TCN effect using HL-60 cells, it is better to present TCN+ATRA combination effect for Western blot, gene expressions analysis, and even morphological study in both cell lines. In that case, it may be better to add microarray analysis using HL-60 cells, to see the difference between NB4 cells.

Originally, the main focus of this paper is about the identification of TCN through the screening of kinase inhibitor library, and clarifying the mechanisms of TCN effect by microarray, by using NB4 APL cells. To follow reviewers’ comments at 1st revision, we compared the effect of TCN and ATRA, revealed the sufficient effect of TCN on non-APL cell line, HL-60 AML cells. The query about the mechanism of TCN for HL-60 AML cells is raised from this 2nd revision, but to answer this, we think that this needs to be thoroughly investigated, which will surely take some time. Indeed, we are now investigating this TCN effect, combined with ATRA, in many AML cells. We started to examine for signaling and gene expressions to reveal the mechanisms, because we also think it is important and to increase the relevance of this series of research.

It would be grateful if you allow as to move forward at this time, and clarify this as a next project.

Lines 246-248: “We further examined the combination effect of TCN and ATRA for the induction of CD11b expression and found that there was no enhancement effect (data not shown), may also suggest the difference of the mechanisms of these agents.” Why not show the data as supporting information? It is important to show that these drugs with pro-differentiation effects together do not increase cell differentiation in the cell lines tested as an available data. After adding the data, please remove “data not shown” from the discussion section (line 324).

We agree with this suggestion. We added this data in Fig S1. 

Following these comments, we feel the improvement of our paper. We appreciate very much for these comments.

---

## [Decision Letter · Decision Letter 2]

25 Apr 2024

Identification of triciribine as a novel myeloid cell differentiation inducer

PONE-D-23-22524R2

Dear Dr. Shinichiro Takahashi,

We’re pleased to inform you that your manuscript has been judged scientifically suitable for publication and will be formally accepted for publication once it meets all outstanding technical requirements.

Kind regards,

Tanay Chaubal

Academic Editor

PLOS ONE

Additional Editor Comments (optional):

Reviewers' comments:

Reviewer's Responses to Questions

**Comments to the Author**

1. If the authors have adequately addressed your comments raised in a previous round of review and you feel that this manuscript is now acceptable for publication, you may indicate that here to bypass the “Comments to the Author” section, enter your conflict of interest statement in the “Confidential to Editor” section, and submit your "Accept" recommendation.

Reviewer #2: All comments have been addressed

2. Is the manuscript technically sound, and do the data support the conclusions?

Reviewer #2: Yes

3. Has the statistical analysis been performed appropriately and rigorously? 

Reviewer #2: Yes

4. Have the authors made all data underlying the findings in their manuscript fully available?

Reviewer #2: Yes

5. Is the manuscript presented in an intelligible fashion and written in standard English?

Reviewer #2: Yes

6. Review Comments to the Author

Reviewer #2: The authors have addressed my previous comments and I look forward to seeing additional results on the effects of TCN in combination with ATRA in multiple AML lineages in future studies from this research group.

7. PLOS authors have the option to publish the peer review history of their article (what does this mean?). If published, this will include your full peer review and any attached files.

Reviewer #2: No

---

## [Editor Report · Acceptance letter]

1 May 2024

PONE-D-23-22524R2 

PLOS ONE

Dear Dr. Takahashi, 

I'm pleased to inform you that your manuscript has been deemed suitable for publication in PLOS ONE. Congratulations! Your manuscript is now being handed over to our production team.

Kind regards, 

on behalf of

Dr. Tanay Chaubal 

Academic Editor

PLOS ONE